# Stability of the COVID-19 At-Home Test after Exposure to Extreme Temperatures

Erin Gick,[a] Stephen McCune,[a] ![ORCID] Jamie P. Deeter[a]

[a]Roche Diagnostics Corporation, Indianapolis, Indiana, USA

**ABSTRACT** To ensure sufficient sensitivity and specificity of lateral flow tests for the detection of SARS-CoV-2 antigen, manufacturers recommend appropriate conditions for storage, including a temperature range. However, there is a high likelihood that kits will be exposed to temperatures outside of this range during transit to some regions. In this prospective study, we evaluated the sensitivity and specificity of the COVID-19 At-Home Test kits (manufactured by SD Biosensor/distributed by Roche) currently being delivered through a US Government program, after exposure to a range of hot and cold temperatures. COVID-19 At-Home Test kits were stored at up to 5 different temperatures: frozen ($-4.0°F$ [$-20.0°C$]), refrigerated ($42.8°F$ [$6.0°C$]), room temperature ($68.0°F$ [$20.0°C$]), warm ($98.0°F$ [$36.7°C$]), and excessive heat (118.0 to 126.0°F [47.8 to 52.2°C]) for 24 h and left at room temperature for 60, 90, or 120 min before use. Test kits were also stored for 48 h, 1 week, or 2 weeks in frozen, warm, and excessive heat conditions, and left for 60 or 120 min before use. In each scenario (storage temperature + time at room temperature), 5 positive and 5 negative control samples were applied, and line intensity was recorded using a color scale (0 to 100%). In every scenario, every positive sample resulted in strong signal intensity ($\geq$26%), and every negative sample returned a negative result. This study suggests that exposure of up to 2 weeks to extreme temperatures, such as those that may occur in transit, does not impact the stability of the COVID-19 At-Home Test.

**IMPORTANCE** COVID-19 At-Home Test kits may be exposed to extreme temperatures in transit, which may impact test sensitivity and specificity. We investigated assay ability to identify SARS-CoV-2 antigen after 24 h to 2 weeks in frozen, refrigerated, room temperature, warm, or excessive heat conditions. The assay correctly identified all positive and negative samples in all scenarios. This study suggests that exposure of up to 2 weeks to extreme temperatures, such as those that may occur in transit, does not impact the stability of the COVID-19 At-Home Test.

**KEYWORDS** SARS-CoV-2, point-of-care testing, extreme heat, freezing

The availability of mass testing and ability of individuals to self-administer tests to detect severe acute respiratory syndrome-related coronavirus 2 (SARS-CoV-2) remains an essential part of tracking and controlling the spread of coronavirus disease 19 (COVID-19) (1). Providing users with information on their disease status allows them to take appropriate action, such as isolating, and seek appropriate medical support.

Lateral flow tests (LFTs) based on SARS-CoV-2 antigen (Ag) have become an affordable and scalable method of providing mass surveillance and testing for an individual's viral status (1–3). The reliability of such testing devices is crucial for providing rapid health information to the user and maintaining public trust in their value. Consequently, they must be able to maintain performance in a variety of handling and storage conditions, including varying temperatures and climates (3).

Through a US Government initiative, free COVID-19 At-Home Test LFT kits (manufactured by SD Biosensor/distributed by Roche) are available for shipping to US citizens who register to receive the kits, at no charge. The instructions for use state that kits should be stored at

Address correspondence to Jamie P. Deeter, jamie.phillips.jp1@roche.com.

The authors declare a conflict of interest. Erin Gick, Stephen McCune, and Jamie P. Deeter are employees of Roche Diagnostics Corporation. Jamie P. Deeter is also a recipient of stock options in Roche Diagnostics Corporation.

**TABLE 1** Test conditions for temperature evaluation of the COVID-19 At-Home Test at 24 h

| Storage temp (°F/°C) | Scenario | No. tests at room temp for 60 min | No. tests at room temp for 90 min | No. tests at room temp for 120 min |
|---|---|---|---|---|
| −4.0/−20.0 | Frozen | 5 positive<br>5 negative | 5 positive<br>5 negative | 5 positive<br>5 negative |
| 42.8/6.0 | Refrigerated | 5 positive<br>5 negative | 5 positive<br>5 negative | 5 positive<br>5 negative |
| 68.0/20.0 | Room temp | 5 positive<br>5 negative | 5 positive<br>5 negative | 5 positive<br>5 negative |
| 98.0/36.7 | Warm | 5 positive<br>5 negative | 5 positive<br>5 negative | 5 positive<br>5 negative |
| 118.0 to 126.0/47.8 to 52.2 | Excessive heat | 5 positive<br>5 negative | 5 positive<br>5 negative | 5 positive<br>5 negative |

36 to 86°F (2 to 30°C), protected from direct sunlight, and should not be frozen (4). These tests are mailed with no temperature control, and can potentially be exposed to extreme temperatures. Consumers may also travel with kits, so the temperature tolerance of the tests is an important parameter to consider (5).

Although the effects of extreme temperatures on the COVID-19 At-Home Test have not previously been evaluated, data from other LFTs offer mixed evidence on performance outside recommended ranges. A study of 11 commercially available SARS-CoV-2 LFTs found that storage for 3 weeks at elevated temperatures (99°F [37°C]) was commonly found to impair sensitivity, whereas 3 days at low temperatures (36 to 39°F [2 to 4°C]) impaired specificity for some LFTs (6). However, short-term (up to 16 h) storage before testing with the Panbio COVID-19 Ag Rapid Test Device (Abbott) at extreme temperatures (freezing at −4.0 to 113.0°F [−20.0 to 45.0°C]) did not impact sensitivity or specificity (7). Cvak et al. (8) have also reported that an aflatoxin LFT maintained performance in Sub-Saharan conditions (≤101.1°F [≤38.4°C], ≤91% relative humidity) at 3 cutoffs throughout a 5-month study period. Focusing on device design, Kaur et al. 2021 (9) evaluated the effects of long-term storage on the nitrocellulose membranes used in LFTs, finding that lateral rates of membranes in sealed packaging were significantly reduced at temperatures of 99°F (37°C) and 113°F (45°C) after 2 months of storage, which concurs with the reduced sensitivity described above.

In addition to storage within recommended temperatures, the FDA recommends leaving test kits unopened at room temperature for ≥2 h before opening to ensure appropriate performance (10). However, if control lines appear as described in the instructions, the test is assumed to be performing appropriately (4).

As the COVID-19 At-Home Test is likely to experience temperatures outside manufacturer recommendations during transit, this study evaluated test stability when exposed to extreme hot and cold conditions during storage, and at multiple time intervals after exposure.

To assess stability after 24 h, COVID-19 At-Home Test kits were stacked on top of each other for 24 h at 5 temperatures: frozen (−4.0°F [−20.0°C]), refrigerated (42.8°F [6.0°C]), room temperature (68.0°F [20.0°C]), warm (98.0°F [36.7°C]), and excessive heat (118.0 to 126.0°F [47.8 to 52.2°C]). Kits were removed from the incubator/freezer/fridge and left at room temperature for 60, 90, or 120 min before use. To assess long-term periods at extreme conditions, kits were stacked on top of each other for either 48 h, 1 week, or 2 weeks at 3 temperatures: frozen (−4.0°F [−20.0°C]), warm (98.0°F [36.7°C]), and excessive heat (118.0 to 126.0°F [47.8 to 52.2°C]). For long-term periods (≥48 h), kits were left at room temperature for 60 or 120 min before use. Five positive and 5 negative controls were evaluated per scenario (temperature + time at room temperature) (Tables 1 and 2).

All tests were conducted using positive and negative BD Veritor System SARS-CoV-2 control swabs (catalog 256087).

As per the assay quick reference instructions (4), samples were added to an extraction buffer tube and 4 drops applied to the sample hole of the assay. After 20 min, visual results were recorded by 2 independent evaluators using daylight lamps. Assay results were based on the color scale for the COVID-19 At-Home Test (Fig. 1), and positive intensity was defined

**TABLE 2** Test conditions for temperature evaluation of the COVID-19 At-Home Test at 48 h, 1 week, and 2 weeks

| Storage temp (°F/°C) | Scenario | No. tests at room temp for 60 min | No. tests at room temp for 120 min |
|---|---|---|---|
| −4.0/−20.0 | Frozen | 5 positive<br>5 negative | 5 positive<br>5 negative |
| 98.0/36.7 | Warm | 5 positive<br>5 negative | 5 positive<br>5 negative |
| 118.0 to 126.0/47.8 to 52.2 | Excessive heat | 5 positive<br>5 negative | 5 positive<br>5 negative |

as strong ($\geq$26%), medium (12 to <26%), weak (>1 to <12%), or very weak (1%). Background clearance scores and flow ratings were recorded to verify assay functionality, and we planned to repeat invalid results as necessary.

The influence of temperature stressing was also checked by testing the functionality of the test kits. All components (test strip, extraction buffer, swabs, and nozzle caps) were stored together, and complete assays tested as per manufacturer instructions.

Overall, 330 assay tests were conducted. At every storage temperature and subsequent time left at room temperature, all 165 positive samples had a strong positive intensity ($\geq$ 26%), and all 165 negative samples gave a negative result with a visible control line. No invalid results occurred during testing, and all assays passed background clearance and flow-rating checks. The functionality of the kits was not impacted by storage conditions.

These highly consistent results, using well-characterized samples for SARS-CoV-2, indicate that COVID-19 At-Home tests are accurate after exposure to extreme temperature conditions for up to 2 weeks. Manufacturer recommendations should always be followed, though we demonstrate that, in cases where kits encounter extreme temperatures for short periods (e.g., in transit) or longer periods (e.g., in mailboxes while households are away), their ability to detect SARS-CoV-2 is unlikely to be impaired.

Although FDA guidance suggests that tests should be left unopened at room temperature for $\geq$2 h (10), our study also suggests that the test's ability to detect SARS-CoV-2 is not compromised by use at shorter durations.

This study does not exclude the possibility that sensitivity and specificity may be reduced after periods longer than 2 weeks, as reported for some commercially available tests (6). The consequences of this may include false-negative test results at clinically relevant virus concentrations compatible with transmission or false-positive results, leading to unnecessary isolation. However, the COVID-19 At-Home Test is not directly comparable, as it uses a different buffer and optimized antibodies.

Overall, although these data indicate that the COVID-19 At-Home Test is suitably robust for postal delivery, there were several limitations. Test kits may spend longer periods than 2 weeks at extreme temperatures or encounter large fluctuations in temperature. There may also be other storage conditions during transport, including exposure to light, dust, or movement, which were not investigated. Additionally, the control samples used may not fully reflect the heterogeneity of samples (and viral loads) encountered in real-world use. Finally, this study only evaluated 1 rapid LFT point-of-care test kit, so we are unable to apply these results to other commercial tests.

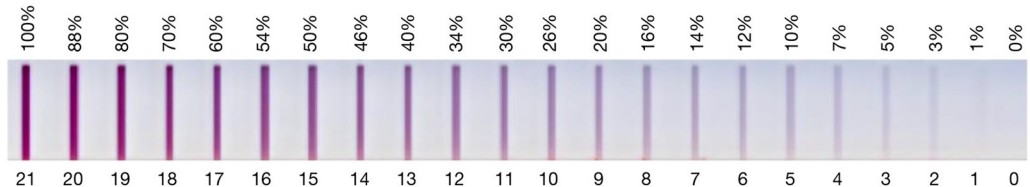

**FIG 1** Color scale for interpreting COVID-19 At-Home Test results.

In conclusion, this study demonstrates that the COVID-19 At-Home Test has robust performance at extreme temperatures outside manufacturer recommendations. Although tests should be kept within recommended limits whenever possible, our data suggest that exposure to extreme temperatures for up to 2 weeks is unlikely to impair the sensitivity or specificity of this assay. This provides additional confidence that tests used outside the recommended temperature range provide accurate results for users, so that they can take appropriate action.

## ACKNOWLEDGMENTS

We thank Simon Lott of Elements Communications Limited (Westerham, UK) for editorial assistance with the preparation of this manuscript. We also thank Priyanka Uprety and Allison McMullen of Roche Diagnostics Corporation for their review.

All product names and trademarks are the property of their respective owners.

Erin Gick, Stephen McCune, and Jamie P. Deeter are employees of Roche Diagnostics Corporation. Jamie P. Deeter is also a recipient of stock options in Roche Diagnostics Corporation.

Funding for medical writing assistance was provided by Roche Diagnostics Corporation.

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
