## [Reviewer comments · Microbiology Spectrum]

Microbiology Spectrum

Stability of the COVID-19 At-Home Test after exposure to extreme temperatures

Jamie Phillips Deeter, Stephen McCune, and Erin Gick

Corresponding Author(s): Jamie Phillips Deeter, Roche Diagnostics Corporation

Review Timeline:

Submission Date:	October 28, 2022
Editorial Decision:	November 22, 2022
Revision Received:	December 12, 2022
Accepted:	December 18, 2022

Editor: Heba Mostafa

Reviewer(s): Disclosure of reviewer identity is with reference to reviewer comments included in decision letter(s). The following individuals involved in review of your submission have agreed to reveal their identity: Olivier Vandenberg (Reviewer #2)

Transaction Report:

DOI: <https://doi.org/10.1128/spectrum.04291-22>

November 22, 2022

Dr. Jamie Phillips Deeter
Roche Diagnostics Corporation
9115 Hague Road
Indiana

Re: Spectrum04291-22 (Stability of the COVID-19 At-Home Test after exposure to extreme temperatures)

Dear Dr. Jamie Phillips Deeter:

Link Not Available

Sincerely,

Heba Mostafa

Journals Department
Reviewer comments:

Reviewer #1 (Comments for the Author):

This is a well written paper demonstrating that once SARS-CoV-2 rapid antigen test is very robust when it comes to exposure to extreme temperatures. The only thing that could be improved are the tables. A summary of the distribution of values for the positive results for the 5 positive samples would be very helpful.

Reviewer #2 (Comments for the Author):

Authors aim to demonstrate that the COVID-19 At-Home Test has robust performance at extreme temperatures outside manufacturer recommendations. Overall, the paper is well written, and the authors have reached their objectives. The presented

data suggest that exposure to extreme temperatures for up to 2 weeks is unlikely to impair the sensitivity or specificity of this assay. However, the authors have only evaluated the performance of only one Lateral Flow Assay (the COVID-19 At-Home Test kits (manufactured by SD Biosensor/distributed by Roche)) currently being delivered through a US Government program).

This represents a major weakness. The lack of evaluation of other rapid diagnostic tests makes it impossible to extend the study findings to other commercial tests. This point should be clearly emphasized by the authors to avoid a false sense of security on the part of health care users and the public.

The viral load of the positive control sample should also be clarified.

With attention to the above details, the paper is worth to be published.

Staff Comments:

Preparing Revision Guidelines

Please return the manuscript within 60 days; if you cannot complete the modification within this time period, please contact me. If you do not wish to modify the manuscript and prefer to submit it to another journal, please notify me of your decision immediately so that the manuscript may be formally withdrawn from consideration by Microbiology Spectrum.

Jamie P Deeter
Roche Diagnostics Corporation
9115 Hague Road, Indianapolis
Indiana 46250-0457
USA
9 December 2022

Dear Reviewers,

On behalf of my fellow authors, I have great pleasure in resubmitting our research article entitled "Stability of the COVID-19 At-Home Test after exposure to extreme temperatures" following receipt of your peer review comments.

We have included a point-by-point response to the reviewers' comments below. Please note that the page and line numbers referred to relate to the version of the manuscript with revisions tracked.

Thank you very much for your review.

Yours sincerely,

Jamie P Deeter

Reviewer #1

This is a well written paper demonstrating that one SARS-CoV-2 rapid antigen test is very robust when it comes to exposure to extreme temperatures. The only thing that could be improved are the tables. A summary of the distribution of values for the positive results for the 5 positive samples would be very helpful.

Response: Thank you for this positive review. While we agree that it would be helpful to provide a distribution of values, unfortunately we were unable to quantify the positive samples based on the starting material and so were unable to quantify the viral load relative to the color intensity (please see response B to reviewer #2). Given this challenge, color intensity was recorded as per the table below at the time of testing. All positive tests were 3+ by these criteria ($\geq 26\%$ intensity; strong positive). Despite not knowing the specific concentration, these tests nevertheless unanimously demonstrated strong positive signals regardless of the extreme temperatures or durations that they were subjected to.

	Result	Symbol	Description
Reactive	Strong positive intensity	3+	Control line and respective test line visible. Intensity of test line $\geq 26\%$ intensity corresponding to the color scale (figure 1)
	Medium positive intensity	2+	Control line and respective test line visible Intensity of test line $\geq 12\%$ <26% intensity corresponding to the color scale (figure 1)
	Weak positive intensity	1+	Control line and respective test line visible. Intensity of test line $\geq 1\%$ <12% intensity corresponding to the color scale (figure 1)
	Very weak positive intensity	w+	Control line and respective test line visible. Intensity of test line = 1% intensity corresponding to the color scale (figure 1)
Negative		-	Control line visible. Respective test line not visible.
Invalid		/	Unusual background of the test strip, no control line, further reasons

Reviewer #2

Authors aim to demonstrate that the COVID-19 At-Home Test has robust performance at extreme temperatures outside manufacturer recommendations. Overall, the paper is well written, and the authors have reached their objectives. The presented data suggest that exposure to extreme temperatures for up to 2 weeks is unlikely to impair the sensitivity or specificity of this assay. However, the authors have only evaluated the performance of only one Lateral Flow Assay (the COVID-19 At-Home Test kits (manufactured by SD Biosensor/distributed by Roche)) currently being delivered through a US Government

program).

A. This [the evaluation of only one assay] represents a major weakness. The lack of evaluation of other rapid diagnostic tests makes it impossible to extend the study findings to other commercial tests. This point should be clearly emphasized by the authors to avoid a false sense of security on the part of health care users and the public.

Response: Thank you, we have added the following statement to the discussion acknowledging this limitation: "Finally, this study only evaluated one rapid LFT point-of-care test kit, so we are unable to apply these results to other commercial tests."

B. The viral load of the positive control sample should also be clarified.

Response: Thank you, this is a fair comment. Per the instructions for use (IFU) for the BD Veritor SARS-CoV-2 Control swab set, the positive "control swabs contain non-infectious, recombinant viral protein antigen with less than 0.1% sodium azide as a preservative." There is no information about viral load, and this information does not appear to be readily available. Unfortunately, when we tried to use contrived, quantified SARS material from SeraCare, the antigen components did not produce a result on the LFT. We also investigated the use of SARS-CoV-2 positive control swabs from Microbiologics; however we did not use the swabs in our status although we did not end up using their swabs in our study as the IFU states "Microbiologics guarantees each nucleic acid is present and can be amplified, and that surface proteins are detectable by an antigen assay *but does not guarantee specific analyte concentrations.*" Ultimately, we are unable to provide a source of positive control swabs that could guarantee a titer or concentration range.

C. With attention to the above details, the paper is worth to be published.

Response: Thank you, we appreciate your positive review.

December 18, 2022

Dr. Jamie Phillips Deeter
Roche Diagnostics Corporation
9115 Hague Road
Indiana

Re: Spectrum04291-22R1 (Stability of the COVID-19 At-Home Test after exposure to extreme temperatures)

Dear Dr. Jamie Phillips Deeter:

Your manuscript has been accepted, and I am forwarding it to the ASM Journals Department for publication. You will be notified when your proofs are ready to be viewed.

Sincerely,

Heba Mostafa
Editor, Microbiology Spectrum
